# USHER: Unbiased Sampling for Hindsight Experience Replay

**Liam Schramm**
Department of Computer Science
Rutgers University
New Brunswick, NJ 08904
lbs105@rutgers.edu

**Yunfu Deng**
Department of Computer Science
Rutgers University
New Brunswick, NJ 08904
yd275@scarletmail.rutgers.edu

**Edgar Granados**
Department of Computer Science
Rutgers University
New Brunswick, NJ 08904
gary.granados@gmail.com

**Abdeslam Boularias**
Department of Computer Science
Rutgers University
New Brunswick, NJ 08904
ab1544@rutgers.edu

**Abstract:** Dealing with sparse rewards is a long-standing challenge in reinforcement learning (RL). Hindsight Experience Replay (HER) addresses this problem by reusing failed trajectories for one goal as successful trajectories for another. This allows for both a minimum density of reward and for generalization across multiple goals. However, this strategy is known to result in a biased value function, as the update rule underestimates the likelihood of bad outcomes in a stochastic environment. We propose an asymptotically unbiased importance-sampling-based algorithm to address this problem without sacrificing performance on deterministic environments. We show its effectiveness on a range of robotic systems, including challenging high dimensional stochastic environments.

**Keywords:** Reinforcement Learning, Multi-goal reinforcement learning

## 1 Introduction

In recent years, model-free reinforcement learning (RL) has become a popular approach in robotics. In particular, these methods stand out in their ability to learn near-optimal policies in high-dimensional spaces [1, 2, 3]. One popular extension of RL, *multi-goal RL*, allows trained robots to generalize to new tasks by conditioning on a goal parameter that determines the reward function. However, RL algorithms often struggle with tasks that involve sparse rewards, as these environments can require a very large amount of exploration to discover good solutions. Hindsight Experience Replay (HER) offers a solution to the sparse reward problem for multi-goal reinforcement learning [4].

HER treats failed attempts to reach one goal as successful attempts to reach another goal. This significantly reduces the difficulty of the exploration problem, because it guarantees a minimum density of reward and ensures that every trajectory receives useful feedback on how to reach some goal, even when the reward signal is sparse. However, these benefits come with a trade-off. While HER is unbiased in deterministic environments, it is known to be asymptotically biased in stochastic environments [5, 6]. This is because HER suffers from a *survivorship bias*. Since failed trajectories to one goal are treated as successful trajectories to another, it follows that HER only ever sees successful trajectories. If a random event can prevent the robot from reaching a desired goal $g$, then HER will only

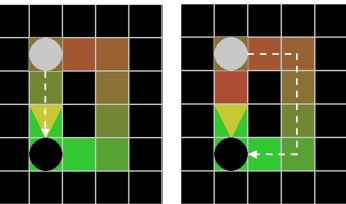

Figure 1: Q-values learned with HER (left), and Q-learning (right). A robot must navigate from the white circle to the black circle while avoiding obstacles (black squares) and risky areas (yellow triangle, 75% chance of stopping the robot). The value function ranges from 1 (bright green) to 0 (bright red).

6th Conference on Robot Learning (CoRL 2022), Auckland, New Zealand.

sample $g$ as a goal when the event did not occur, leading it to
significantly overestimate the likelihood of success and underestimate the likelihood of dangerous
events. Practically, this manifests as a tendency for HER to want to "run red lights" and take risks.

We present a concrete toy example of this problem in Figure 1, using tabular Q-learning. As we can
see, HER values the direct path to the goal and the square en route to the dangerous square much
higher than that path's correct Q-value because it underestimates the risk. HER learns to take the
shorter, more dangerous path and achieves a lower success rate with lower reward than Q-learning.

As suggested in both [5] and [6], we derive an approach that allows us to use HER for sampling
goals without suffering from these bias problems. We do this by separating the goal used for the
reward function ($g_r$) from the goal that is passed to the policy ($g_\pi$). The value function is conditioned
on both goals, but only the reward goal is sampled using HER. This allows us to efficiently learn a
successor representation over future achieved goals that we can use for importance sampling. We
show that reweighting HER's mean squared Bellman error using this successor representation yields
an unbiased estimate of the error. We call this method Unbiased Sampling for Hindsight Experience
Replay (USHER). We demonstrate this approach on an array of stochastic environments, and find that
it counteracts the bias shown by HER without compromising HER's sample efficiency or stability.

## 2    Definitions

We define a *multi-goal* Markov Decision Process (MDP) as a seven-tuple: state space $S \subseteq \mathbb{R}^n$, action
space $A \subseteq \mathbb{R}^m$, discount factor $\gamma \in [0, 1]$, transition probability distribution $P(s' \mid s, a)$ (with density
function $f(s' \mid s, a)$) for $(s, a, s') \in S \times A \times S$, goal space $G \subseteq \mathbb{R}^l$, goal function $\phi : S \to G$, and
reward function $R : S \times G \to \mathbb{R}$. A goal $g = \phi(s) \in G$ is a vector of goal-relevant features of state
$s \in S$. Goal function $\phi$ is defined a priori, depending on the task. A typical example of $\phi(s)$ is a
low-dimensional vector that preserves only the entries of state-vector $s$ that are relevant to the goal.
For instance, a mobile robot is tasked with moving to a particular location and arriving there at zero
velocity. The state space of the robot would include velocities and orientations of each wheel, along
with several other attributes that are needed to control the robot. The goal function would take the
full high-dimensional state of the robot and return only its location and velocity. Therefore, each goal
point corresponds to a subspace of the state space in this example. A special case is when $G = S$
and $g = \phi(s), \forall s \in S$. Note that the immediate reward function $R(s, g)$ depends on a selected goal
$g \in G$. Every selection of $g \in G$ produces a valid single-goal MDP. We denote by $\pi$ a deterministic
goal-conditioned policy, with $\pi(s, g) \in A$ for $s \in S, g \in G$, and define $Q^*(s, a, g)$ to be the unique
optimal Q-value of action $a \in A$ in state $s \in S$, given selected goal $g \in G$.

In the proposed algorithm and analysis, a policy $\pi$ can be evaluated according to a goal that is not
necessarily the same goal used by the policy for selecting actions. Therefore, we use $g_\pi$ to refer to
goals that are passed to policies, and $g_r$ to denote goals that are used to evaluate policies. Using these
notations, the Bellman equation is re-written as

$$Q^\pi(s, a, g_r, g_\pi) = \mathbb{E}_{s'}[R(s', g_r) + \gamma Q^\pi(s', \pi(s', g_\pi), g_r, g_\pi) \mid s, a].$$

Intuitively, this means "The expected cumulative discounted sum of rewards $R(s', g_r)$, when using
policy $\pi(s', g_\pi)$". The reason for this separation is that it allows us to more easily separate the
problem of predicting future rewards from the problem of directing the policy. This makes it much
easier to find an analytic expression for HER's bias. In particular, it lets us learn an expression for
future goal occupancy that is conditioned only on $g_\pi$ and not $g_r$, which will allow us to correct
for the bias induced by hindsight sampling. Observe that when $g_r = g_\pi$, this definition reduces
to the Bellman equation for standard multi-goal RL. For standard Q-learning, $\pi(s', g_\pi)$ would be
$\arg\max_{a'} Q(s', a', g_\pi, g_\pi)$, where both the policy and reward goals are set to $g_\pi$.

**HER.** HER is a modification of the experience replay method employed by many deep RL algo-
rithms [1, 4, 7, 8, 9]. Policy goal $g_\pi$ is sampled before each trajectory begins, and is not changed
while generating the trajectory. After generating a trajectory, HER stores the entire trajectory in the
replay buffer. When sampling transitions $(s, g_\pi, a, s')$ from the buffer, HER retains the original goal

$g_\pi$ used in the policy that generated the trajectory, i.e., $g_r \leftarrow g_\pi$, with probability $\frac{1}{k+1}$, where $k$ is a natural number (usually 4 or 8). The rest of the time, it replaces the original goal with $\phi(s_t)$, i.e., $g_r \leftarrow \phi(s_t)$, where $s_t$ is a randomly sampled state from the future trajectory that starts at $s$. Goals that are selected from the future trajectory are referred to as "hindsight goals". HER then updates the Q-value and policy networks with $(s, g_\pi, a, s', R(s', g_r))$.

## 3 Related Work

Over the last few years, several methods have attempted to address the hindsight bias induced by HER. ARCHER attempts to decrease HER's hindsight bias by multiplying the loss on hindsight goals and non-hindsight goals by different weights, effectively upweighting the importance of hindsight goals [10]. MHER extends HER to a multi-step RL and proposes a bias/variance tradeoff for that setting [11]. A rigorous mathematical approach to HER's hindsight bias is taken in [5], by showing that HER is unbiased in deterministic environments, and that one of HER's key benefits is ensuring a minimum density of feedback from the reward function, even in high-dimensional spaces where the reward density would normally be extremely low. This reward-density problem is addressed by deriving a family of algorithms (called the $\delta$-*family*, e.g. $\delta$-DQN, $\delta$-PPO), which guarantees a minimum reward density while still being unbiased. These methods do not use HER and have higher variance. The authors of [5] also state that the problem of formulating an unbiased form of HER is still open, and call for additional research into the problem.

Bias-Corrected HER (BHER) attempts to account for hindsight bias by analytically calculating importance-sampling hindsight goals [12]. Unfortunately, we believe that this derivation is incorrect. The proof in BHER relies on the assumption that the probability of a transition is independent of the goal ($f(s' \mid s, a, g) = f(s' \mid s, a)$). This assumption does not hold for HER, because it samples the goal from the future trajectory of $s$, which depends on $s'$. Both our work and [5] give concrete counterexamples to this assumption. The following derivation provides an unbiased solution that does not rely on this flawed assumption.

## 4 Derivation

**Bias in HER.** We derive the formula of the bias introduced by HER in estimating the Q-value function in the following. Let $s, a$, and $s'$ be random variables representing a state, action, and subsequent state in a given trajectory generated by policy $\pi$ with goal $g_\pi$. Let $T$ be the number of time-steps remaining in the sub-trajectory that starts at $s$. Let $Q^\pi_{HER}(s, a, g_r, g_\pi)$ be the solution to the Bellman equation obtained using HER's sampling process of reward goal $g_r$ (Sec. 2). This sampling process takes into account both $g_\pi$ and $T$. Furthermore, $g_r$ is selected from the sub-trajectory that starts at $s$ with probability $\frac{k}{k+1}$. Therefore, the probability $f(s' \mid s, a, g_r, g_\pi, T)$ of the next state $s'$ after knowing $g_\pi, g_r$ and $T$ is generally not the same as $f(s' \mid s, a)$, which is what HER uses empirically to estimate $Q^\pi_{HER}(s, a, g_r, g_\pi)$. The following proposition quantifies this bias ratio.

**Proposition 1.** *Suppose $g_\pi$ is fixed at the start of the trajectory, and $g_r$ is sampled using HER. Then for any $s', s, a, g_r, g_\pi, T, f(s' \mid s, a, g_r, g_\pi, T) = \frac{f(g_r \mid s', \pi(s', g_\pi), g_\pi, T-1)}{f(g_r \mid s, a, g_\pi, T)} f(s' \mid s, a)$.*

**Proof:** Appendix (A.5). This identity presents an interesting corollary.

**Corollary.** *Suppose $Q^\pi_{HER}(s, a, g_\pi, g_\pi)$ satisfies the Bellman equation and the distribution of future achieved goals is absolutely continuous with respect to the goal space for all $s, a, g_\pi$, and $\pi(s, g_\pi) = \arg\max_{a'} Q^\pi_{HER}(s, a', g_\pi, g_\pi)$. Then $Q^\pi_{HER}(s, a, g_\pi, g_\pi) = Q^*(s, a, g_\pi)$, where $Q^*$ is the optimal goal-conditioned Q-function.*

**Proof:** Appendix (A.6). While this establishes that the target value for $Q^\pi_{HER}$ is unbiased when $g_r = g_\pi$, the function approximator for $Q^\pi_{HER}$ may still be biased, because values other than $g_r = g_\pi$ may influence it through the training of the network. Thus, it is possible that the learned $Q^\pi_{HER}$ value may remain biased until unacceptably large amounts of data are gathered. Additionally, since the density of data is discontinuous, $Q^\pi_{HER}$ may be discontinuous and difficult to approximate with a

neural network. The rest of this section is devoted to developing an importance sampling method that is guaranteed to be asymptotically unbiased over the entire domain of $Q$.

**Unbiased HER.** To estimate $Q^\pi(s, a, g_r, g_\pi)$, the solution to the unbiased Bellman equation, we use in this work the following expression,

$$Q^\pi(s, a, g_r, g_\pi) = \mathbb{E}_{s'}[M(s', s, a, g_r, g_\pi, T)\big(R(s', g_r) + \gamma Q^\pi(s', \pi(s', g_\pi), g_r, g_\pi)\big) \mid s, a, g_r, g_\pi, T],$$

where $M(s', s, a, g_r, g_\pi, T)$ is a weight that cancels the bias ratio given in Proposition 1. Conditioning the expected value over $s'$ on $g_r, g_\pi$, and $T$ frees us from the constraint that $s'$ needs to be independent of $g_r, g_\pi$, and $T$. This would allow us to select $g_r$ from the future trajectory of $s$, as HER does. Note that conditioning on $T$, the number of steps left in the trajectory, is necessary because the distribution of goals selected by HER is not time-independent.

Proposition 1 is useful for understanding what situations may cause HER to be biased, but unfortunately we cannot directly use it for importance sampling. Weighting samples by setting $M(s', s, a, g_r, g_\pi, T)$ as $\frac{f(g_r|s,a,g_\pi,T)}{f(g_r|s',\pi(s',g_\pi),g_\pi,T-1)}$ would require $f(g_r \mid s', \pi(s', g_\pi), g_\pi, T - 1)$ to always be greater than 0, which is not necessarily true. To solve this, we sample a mixture of hindsight goals and goals drawn uniformly from the goal space $G$. Of the goals where $g_r \neq g_\pi$, a fraction $\alpha$ of our goals will be drawn uniformly from the goal space, and the remaining $1 - \alpha$ will be drawn from the trajectory that follows $s$. This results in the following identity,

**Proposition 2.** *Let* $W(s', s, a, g_r, g_\pi, T) = \frac{f(g_r|s,a,g_\pi,T)}{\alpha f(g_r|s,a,g_\pi,T)+(1-\alpha)f(g_r|s',\pi(s',g_\pi),g_\pi,T-1)}$. *Let* $\alpha$ *be a real value in the range* $(0, 1]$. *Then for any* $s', s, a, g_r, g_\pi$,

$$f(s' \mid s, a) = W(s', s, a, g_r, g_\pi, T)\big(\alpha f(s' \mid s, a) + (1 - \alpha)f(s' \mid s, a, g_\pi, g_r, T)\big)$$

*Furthermore, for any function $F$ of state $s'$,*

$$\begin{aligned}
\mathbb{E}_{s'}[F(s') \mid s, a] = \; & \alpha \mathbb{E}_{s'}[W(s', s, a, g_r, g_\pi, T)F(s') \mid s, a] \\
& + (1 - \alpha)\mathbb{E}_{s'}[W(s', s, a, g_r, g_\pi, T)F(s') \mid s, a, g_\pi, g_r, T].
\end{aligned} \tag{1}$$

**Proof:** Appendix (A.7). We can now derive an unbiased variant of HER by applying Proposition 2 to Bellman equation.

**Corollary.** *Suppose $\pi$ is a deterministic policy, $g_r$ is sampled from the previously mentioned mix of hindsight and uniform random goals, and $g_\pi$. Then for any $s', s, a, g_r, g_\pi, T$,*

$$\begin{aligned}
Q(s, a, g_r, g_\pi) = \; & \alpha \mathbb{E}_{s'}[W(s', s, a, g_r, g_\pi, T)\big(R(s', g_r) + \gamma Q(s', \pi(s', g_\pi), g_r, g_\pi)\big) \mid s, a] \\
& + (1 - \alpha)\mathbb{E}_{s'}[W(s', s, a, g_r, g_\pi, T)\big(R(s', g_r) + \gamma Q(s', \pi(s', g_\pi), g_r, g_\pi)\big) \mid s, a, g_\pi, g_r, T]
\end{aligned} \tag{2}$$

This corollary provides us with a simple method of estimating $Q(s, a, g_r, g_\pi)$ using HER. A similar unbiased expression can be derived for estimating the gradient of the Bellman error with respect to the weights of a $Q$-function network, instead of estimating $Q(s, a, g_r, g_\pi)$ directly from samples.

**Learning the future goal distribution.** In order to use the proposed unbiased estimator of the Q-function with policy and reward goals, we need to compute weight $W$ defined in Proposition 2. This can be achieved by learning future goal distributions $f(g_r \mid s, a, g_\pi, T)$ and $f(g_r \mid s', \pi(s', g_\pi), g_\pi, T - 1)$, which both correspond to the conditional probability that a given goal $g_r$ will be selected as a hindsight goal by HER. A technique for learning such long-term distributions, introduced in [5], consists in training a network $f_\theta$, with parameters $\theta$, to approximate the density of future goals $f(g_r \mid s, a)$. The following estimator for the gradient is used in [5], sampling $(s, a, s')$ from transitions, and fixing $g_r$ at the start of each trajectory,

$$\nabla_\theta \big(\mathbb{E}_{s,a}[-f_\theta(s, a, \phi(s))] + \mathbb{E}_{s,a,s',g_r}[f_\theta(s, a, g_r)(f_\theta(s, a, g_r) - \gamma \max_{a'} f_{\text{target}}(s', a', g_r)]\big),$$

wherein $f_{\text{target}}$ is a copy of $f_\theta$ that is updated separately. This method has however a significantly higher variance than HER [5]. We examine here the source of this variance, and explain how separating the policy and reward goals allows us to avoid this variance problem. One issue with this method that can contribute to variance is that the gradient is separated into two parts: one in which

the goal comes from the state ($\phi(s)$), and one in which the goal is sampled at the start of the trajectory ($g_r$). This is a problem, because the gradient at the state-derived goals is strictly negative, while the gradient at the sampled goals is usually positive. In our experiments, this led to a pattern where the value at the state-derived goals would diverge unboundedly, until a goal was sampled sufficiently close to make the value function crash back down to zero, and then the process would repeat again. In other words, it is not guaranteed that $f_\theta$ converges a fixed point for every finite set of trajectories.

One way to avoid this problem would be to have a fixed, non-zero chance that $\phi(s) = g_r$, so that $f_\theta$ always converges to a fixed point given any set of training trajectories. We use HER to achieve this outcome. This is possible, unlike in [5], because we can use the importance sampling method derived above to sample a mixture of HER goals and goals independent of the state. Since HER draws from the future states of $s$, observe that $f(g_r \mid s, a, g_\pi, T)$ is in fact a *successor representation* [13], using an average-reward formulation (because the probability of selecting any of the next T states is uniform). Observe that we can define this probability as

$$f(g_r \mid s, a, g_\pi, T) = \mathbb{E}_{s'}[\frac{1}{T}\delta(g_r - \phi(s')) + (1 - \frac{1}{T})f(g_r \mid s', \pi(s', g_\pi), g_\pi, T - 1) \mid s, a],$$

wherein $\delta$ is Dirac delta function. This results in the loss gradient:

$$\nabla_\theta\Big(\mathbb{E}_{s,a,g_r,g_\pi,T}[-\frac{2}{T}f_\theta(s, a, \phi(s), g_\pi, T) + \mathbb{E}_{s'}[L(s, a, s', g_r, g_\pi, T) \mid s, a]]\Big), \quad (3)$$

$L(s, a, s', g_r, g_\pi, T) \triangleq f_\theta(s, a, g_r, g_\pi, T)\big(f_\theta(s, a, g_r, g_\pi, T) - \gamma f_{\text{target}}(s', \pi(s', g_\pi), g_r, g_\pi, T - 1)\big)$. While $f_\theta$ may not be a true probability density (because it may not integrate to 1), this does not matter for our purposes, as this factor will divide out when we calculate $W$. Finally, we inject the formula in Equation 3 into Equation 1, while replacing $F$ with $f_\theta$, to derive the following unbiased loss gradient,

$$\nabla_\theta\mathcal{L} = \nabla_\theta\mathbb{E}[-\frac{2}{T}f_\theta(s, a, \phi(s), g_\pi, T) + \alpha\mathbb{E}_{s'}[W(s, a, s', g_r, g_\pi, T)L(s, a, s', g_r, g_\pi, T) \mid s, a]$$
$$+ (1 - \alpha)\mathbb{E}[W(s, a, s', g_r, g_\pi, T)L(s, a, s', g_r, g_\pi, T) \mid s, a, g_r, g_\pi, T]]. \quad (4)$$

Note that the values of $\alpha$ we use for learning $Q_\theta$ (Equation 2) and goal distribution densities $f_\theta$ (Equation 4) can be different. For discrete environments, we can learn the future distribution of the goal state using simple tabular methods, such as tabular successor representations.

## 5 Algorithm and Implementation

USHER may be implemented atop DDPG [7], SAC [8], TD3 [9], or any other continuous RL algorithm, as it only changes the loss function for training the goal-conditioned Q-value network. In our experiments, we use SAC as a base. USHER calculates the loss as follows: It samples a batch of transitions $(s, a, s', g_\pi, T)$ from the replay buffer, along with two sets of goals: $g_r$, which is drawn from the future distribution of $s$, and $g'_r$, which is drawn uniformly from the goal space $G$. For each set of goals, we calculate two values of $W$, $W_{\alpha_Q}$ and $W_{\alpha_f}$. We omit the full training loop here, as it is identical to standard HER except for the loss computation. To minimize the variance induced by importance sampling, we clip $W_{\alpha_Q}$ and $W'_{\alpha_Q}$ to the range $[\frac{1}{1+c}, 1 + c]$, where $c$ is a hyperparameter. This allows us to make a bias/variance trade-off between hindsight bias and the variance induced by importance sampling. We find that performance is best for $c \approx 0.3$, and that the bias induced by clipping is negligible for $c > 1.0$ for most environments. We approximate $W$ using $f_\theta$ for all experiments. In order to reduce the total number of neural network evaluations, we made $Q_\theta$ and $f_\theta$ two heads of a two-headed neural network. Although this choice conditions the value function on $T$, the expected gradient for the policy remains the same.

## 6 Experiments

### 6.1 Discrete environment

We first demonstrate our method in the discrete case described in the introduction in Figure 1 because it is analytically tractable and allows us to verify that

**Algorithm 1** Update Rule for USHER

---

**Input:** *Replay Buffer B, Two-headed Critic Network with weights θ, Actor Network with weights w, Weighting Factor $\alpha_Q$ for $Q$, Weighting Factor $\alpha_f$ for $f$, Goal Space G, Goal Function $\phi$;*
*Sample batch of tuples $(s, a, s', g_\pi, T)$ from B;*
*critic_loss = 0; actor_loss = 0*
*Define $W(s, a, s', g_\pi, g_r, T, \alpha) = \frac{f_\theta(g_r|s,a,g_\pi,T)}{\alpha f_\theta(g_r|s,a,g_\pi,T)+(1-\alpha)f_{target}(g_r|s',\pi(s',g_\pi),g_\pi,T-1)}$;*
**for** each *sampled tuple $(s, a, s', g_\pi, T) \in B$* **do**
    *With probability $\frac{k}{k+1}$: $g_r$ =Sample from future trajectory of s; Else: $g_r = g_\pi$*
    *$target\_q = R(\phi(s'), g_r) + Q_{target}(s', \pi(s', g_\pi), g_r, g_\pi, T-1)$; // $Q_{target}$ is a copy of $Q_\theta$*
    *$target\_q' = R(\phi(s'), g_r') + Q_{target}(s', \pi(s', g_\pi), g_r', g_\pi, T-1)$;*
    *$W_{\alpha_f} = (1 - \alpha_f)W(s, a, s', g_\pi, g_r, T, \alpha_f)$; $W'_{\alpha_f} = \alpha_f W(s, a, s', g_\pi, g_r', T, \alpha_f)$*
    *$W_{\alpha_Q} = (1 - \alpha_Q)W(s, a, s', g_\pi, g_r, T, \alpha_Q)$; $W'_{\alpha_Q} = \alpha_Q W(s, a, s', g_\pi, g_r', T, \alpha_Q)$*
    *$critic\_loss \mathrel{-}= \frac{2}{T} f_\theta(\phi(s') \mid s, a, g_\pi, T)$*
    *$critic\_loss \mathrel{+}= W_{\alpha_f}(f_\theta(g_r \mid s, a, g_\pi, T) - f_{target}(g_r \mid s', \pi(s', g_\pi), g_\pi, T-1))^2$*
    *$critic\_loss \mathrel{+}= W'_{\alpha_f}(f_\theta(g_r' \mid s, a, g_\pi, T) - f_{target}(g_r' \mid s', \pi(s', g_\pi), g_\pi, T-1))^2$*
    *$critic\_loss \mathrel{+}= W_{\alpha_Q}(Q_\theta(s, a, g_r, g_\pi, T) - target\_q)^2$*
    *$critic\_loss \mathrel{+}= W'_{\alpha_Q}(Q_\theta(s, a, g_r', g_\pi, T) - target\_q')^2$*
    *$actor\_loss \mathrel{-}= Q_\theta(s, \pi_w(s, g_\pi), g_r = g_\pi, g_\pi, T)$*
**end for**
*Backprop $critic\_loss$ and update $\theta$*
*Backprop $actor\_loss$ and update $w$*

---

USHER learns the correct value function. The environment used has a short, risky path that has a high chance of disabling the robot, and a longer risk-free path. The longer path has a higher expected reward, but we find that HER mistakenly prefers the riskier path. The value functions for USHER and Q-learning both quickly converge to the expected value, while HER overestimates the expected reward.

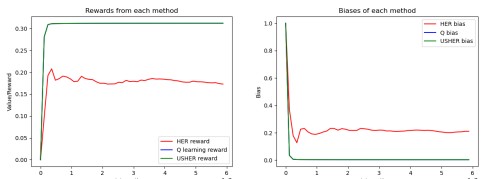

Figure 2: Average reward and bias for HER, USHER, and Q-learning on the long/short path environment.

### 6.2 4-Torus with Freeze

*N-Torus with Freeze* (Fig 3) is a benchmark environment introduced by *Unbiased Methods* that demonstrates HER's bias. Robots navigate a torus with an *N*-dimensional surface to reach a goal. There is also a "Freeze" action, which causes the agent to jump to a random location and then permanently freeze in place and not move again. Further details can be found in [5] or in the Appendix (A.1.2). HER learns to always take the freeze action and fails as a result, while USHER learns a successful policy. DDPG and $\delta$-DDPG are unbiased in this environment, but DDPG struggles due to the difficulty of exploring in high dimensions, and $\delta$-DDPG struggles with its variance.

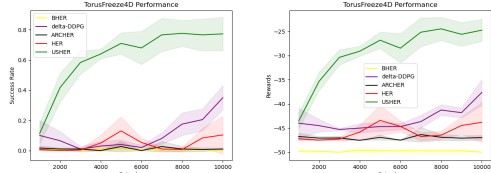

Figure 3: Success Rates (left) and Average Rewards (right) for the 4-Torus with Freeze environment

### 6.3 Car Environment with Random Noise

This environment (Fig 4) uses the "Simple Car" dynamics described in [14]. The robot must navigate around walls while subject to Gaussian action noise [15]. HER performs well for low noise values, but tends to overestimate values more as the noise level rises. USHER suffers significantly less from high noise levels than HER.

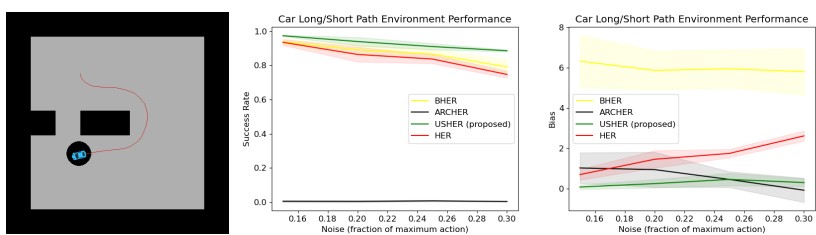

Figure 4: Success Rates (middle) and Bias (right) for the Stochastic Car environment

## 6.4 Red Light Environment

Here, we expand on the example given in the introduction, where HER learns to run red lights because it underestimates the likelihood of a crash. This environment (Fig 5) uses the same car dynamics as the short/long path environment, but change the map to be two sections separated by an intersection and a traffic light. If the car is in the intersection while the traffic light is red, than there is a 75% chance per unit time that the car will be in an accident and break. The green and yellow lights are both safe, and the initial color of the light is random. We find that HER learns to run the red light and immediately attempt to reach the goal, while USHER learns to wait for the red light to end. This results in USHER achieving both higher success rates and higher rewards.

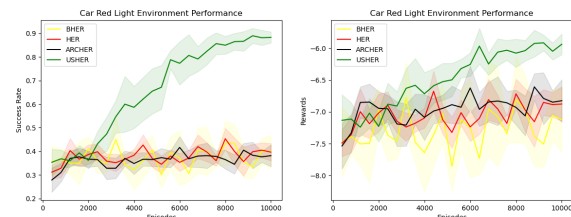

Figure 5: Success rates (left) and average reward (right) for the Red Light Environment

## 6.5 Fetch Robot Environments

While the $\delta$-Actor Critic succeeded at being unbiased, this came at the expense of performance in deterministic environments, due to the method's increased variance [5]. Our goal in this section is to show that USHER does not suffer from this trade-off, and delivers performance competitive with standard HER. To do this, we compare the performance of HER and USHER on several Fetch Robot object manipulation tasks (Fig 6), as these were the tasks HER was originally designed for. These three environments task a robot manipulating a robot arm to reach a point, push an object, and slide an object to a point outside of the robot's reach, respectively. USHER is able to match HER's performance on all of the tested environments (Fig 7). This suggests that the importance sampling method does not significantly affect USHER's variance or sample efficiency in deterministic environments, where HER is known to be unbiased. It also significantly outperforms two other unbiased methods, DDPG and $\delta$-DDPG on FetchReach. Note that although BHER performs slightly better than HER, it takes approximately 10x as long to train as HER, due to needing to evaluate the policy for the entire trajectory for every sampled goal at training time.

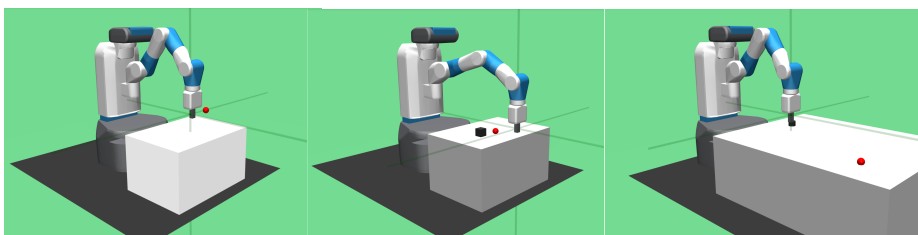

Figure 6: FetchReach (left), FetchPush (middle), and FetchSlide (right) Environments

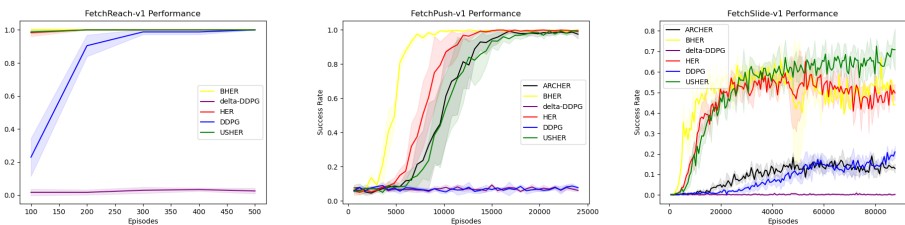

Figure 7: Success Rates for the FetchReach, FetchPush, and FetchSlide Environments

## 6.6 Mobile Throwing Robot

We design a simulated robot arm on a mobile base, and task it with throwing a ball to a randomly selected location (Fig 8). There is also a 50% chance of wind that can blow the ball off course. USHER matches HER's sample efficiency until the point where HER's bias causes its performance to suffer. USHER's performance, by contrast, continues to grow steadily to a 75% success rate, significantly better than HER's 55%. Interestingly, we find that USHER actually underestimates its reward here. This is likely because this environment is slightly non-Markovian, because the wind is sampled at the beginning of each trajectory, and then remains fixed. USHER's proof of unbiasedness, however, assumes that the environment is Markovian. It is interesting to note that USHER still performs well even when this property does not completely hold.

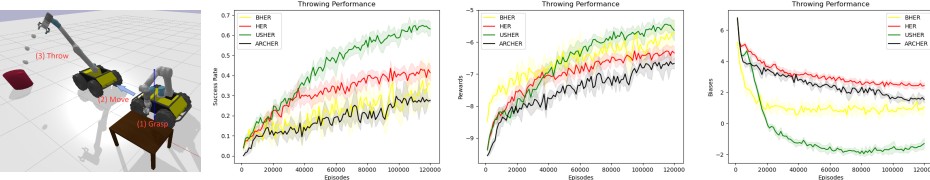

Figure 8: (Left to right) Visualization of the experiment, Success Rates, Rewards, and Bias for the Mobile Robot Throwing experiment

## 6.7 Navigation on a physical mechanum robot

Lastly, we train a mechanum robot to navigate around obstacles to reach a goal and deploy it on a physical robot (Fig 9). The terrain contains a high friction zone that leads to the goal faster, but unreliably. Transfer was done by rolling out trajectories in simulation, and then deploying the same sequence of actions on the physical robot as an open-loop control. We find that USHER outperforms HER. Both robots take the short

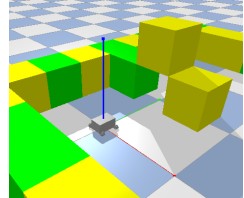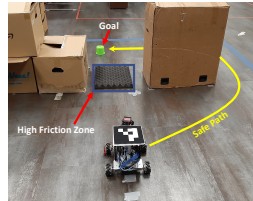

Figure 9: Simulated Mechanum Robot(left) and Physical Mechanum Robot(right)

goal when it is open. When the path is blocked, HER repeatedly slams into the obstacle. By contrast, USHER runs into the block once, and then turns to go around it if it is blocked. This leads USHER to have a higher success rate. In simulation, HER's success rate is approximately 50%, while USHER's is near 100%. Due to the difficulty of transfer, USHER's performance drops on the physical robot, but it still outperforms HER. HER succeeded on 4/10 goals, while USHER succeeds on 6/10. Both methods succeed 100% of the time on the unblocked path environment.

## 7 Limitations

One limitation of this work is that we rely on the Markov assumption to derive our importance sampling weights. This means that while we can correctly estimate the value function for stochastic transitions, we cannot guarantee that the learned value is correct in environments with hidden information. It is unclear whether this is actually an issue in practice, as USHER still outperforms HER on the non-Markovian environments we tested (such as the Throwing Bot). Additionally, USHER requires approximately 2.5 times as many neural net evaluations as HER does per batch update. This was not an issue in our experiments, as the cost of simulation and policy evaluations usually dominated the training time.

## 8 Conclusion

We derive an unbiased importance sampling method for HER, and show that it is able to effectively counteract HER's hindsight bias. We find that addressing this bias leads to higher success rates and rewards in a range of stochastic environments. Furthermore, we introduce a mathematical framework to justify our method which can be used to examine the situations where HER is likely to experience significant bias. In future work, we hope to examine the finite-sample case, in order to better understand whether HER introduces a bias there, and if so, how it could be corrected.

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
