# OpenReview forum: "USHER: Unbiased Sampling for Hindsight Experience Replay"
_robot-learning.org/CoRL/2022/Conference — CoRL 2022 Poster_

### Official Review · Reviewer_mpQg · 2022-07-29

**Originality:** Very Good
**Technical Quality:** Very Good
**Clarity Of Presentation:** Very Good
**Impact:** 4

**Recommendation:**

Weak Accept: I recommend accepting the paper, but will not argue for my recommendation if the majority of other reviewers have a different opinion.

**Summary:**

The paper proposed an algorithm that leverages importance sampling to unbias hindsight experience replay in stochastic environments. This is achieved by involving separate goals for rewards function and for the policy correspondingly. By conditioning both goals to the value function, a successor representation can be learned for importance sampling, where Bellman error using the representation provides unbiased error estimation. Concrete mathematical proof is provided to support the claim the HER is biased and how the algorithm could reach unbiased HER. Sufficient experiments were conducted in simulation to support the algorithm.

**Issues:**

See above

**Quality Of The Limitations Section:**

Limitations are addressed clearly

**Reviewer Expertise:**

3: The reviewer is fairly confident that the evaluation is correct

**Robotics Focus:**

Sufficient demonstration on hardware

**Strengths And Weaknesses:**

Strengths:
- The developed algorithm is novel, challenging and interesting. It involves importance sampling to unbias - HER with concrete mathematical proof to verify the efficacy of the algorithm.
- The paper is well written and easy to follow.
- The design choices are reasonable and has been reflected in the ablation study.
- Comprehensive simulation experiments were conducted to support the paper.

Weaknesses:
Although the simulation experiments are convincing. The Real robot experiment is a bit weak. Only 10 episodes were executed for each of the methods. Given the complexity, noise, uncertainty and domain gap posed by the real environment, 6/10 USHER vs 4/10 HER is far from enough to claim USHER outperforms HER.


**Summary Of Recommendation:**

It is a solid paper that addresses bias in HER. The algorithm is backed by math and the simulation experiments are convinving. The only concern is the real world experiments. The sample size is inadequate to make any conclusions.

---

> ### Author Response · Authors · 2022-08-28
> **Response to Reviewer mpQg**
>
> We thank the reviewer for their feedback and positive review. Due to the time demands of running other experiments, we have not been able to add additional physical robot experiments. However,  we would like to call the reviewers attention to one element of our existing data. USHER always attempted avoid the central obstacle when it was present, and HER never attempted to go around the obstacle. We believe this is a core takeaway from these experiments, and one that is unlikely to change with more data. HER cannot reach the goal without avoiding the obstacle in the center, and we have shown that USHER is at least sometimes successful at reaching the goal while avoiding the obstacle. Both methods always reached the goal when no obstacle was present. We take this to mean that our deployment struggled somewhat with the dead-reckoning element of navigating around the obstacle, but that USHER successfully accounted for the risk posed by the obstacle when deployed on the physical robot.

---

### Official Review · Reviewer_Z2DH · 2022-08-01

**Originality:** Good
**Technical Quality:** Fair
**Clarity Of Presentation:** Fair
**Impact:** 2

**Recommendation:**

Strong Reject: I recommend rejecting the paper and will argue for my recommendation even if other reviewers hold a different opinion.

**Summary:**

The paper proposes a novel method to alleviate bias in hindsight experience replay (HER) using importance sampling. The authors present experiments on a range of environments which suggest that this method boosts performance beyond the original HER algorithm.

**Issues:**

### Larger issues
 * The authors should spend some time reorganizing Section 6 to be easier to follow and give more details about the environments they use for evaluation.
 * The authors should modify Algorithm 1 to be more concise and useful.
 * The authors should add the comparisons to other algorithms they mention in the related works section.

### Small Issues
 * Missing space on line 184
 * Should line 195 say $C < 1.0$?


**Quality Of The Limitations Section:**

Limitations are addressed clearly

**Reviewer Expertise:**

4: The reviewer is confident but not absolutely certain that the evaluation is correct

**Robotics Focus:**

Sufficient demonstration on hardware

**Strengths And Weaknesses:**

# Strengths
 * The method seems interesting, and has potential to be impactful in the research community upon further experimental evaluation.
 * I don't have the theoretical expertise necessary to verify the authors' proofs, but the method seems to be grounded well in theoretical analysis.
 * The authors present results on robot hardware.
 * The abstract and introduction do a good job motivating the problem and introducing the authors' solution.

# Weaknesses
 * A key weakness of the paper is presentation
     * Algorithm 1 is far too crowded to be useful. The authors should replace some of the math with references to the rest of the text.
     * For some reason when the authors show the environments in Figure 2, they only show a subset of them. For example, what is 4-Torus with Freeze? All environments included in the paper should be either explained in the main text or supplement.
     * The results summary section is very disorganized. The authors present all of their results in a gigantic Figure 3 without any helpful captions or references to them in the text. The authors should reorganize this section to be more clear.
 * The authors' core claim is that their proposed method does a better job alleviating HER bias than other work (ARCHER, BHER) that has attempted to do the same in the past, but they never compare to any of this work in experiments. In order to contextualize this work with respect to other literature, these comparisons are critical.

**Summary Of Recommendation:**

Overall, I think the method has promise and encourage the authors to address reviewer concerns during the rebuttal period. I feel that my main concerns about presentation issues and lacking comparisons make the paper unpublishable in its current state, but will happily increase my score if the authors can fully address these issues during the rebuttal period.

The theoretical results seem interesting but I don't have the expertise to evaluate them so I have not considered them while writing my review.

---

> ### Author Response · Authors · 2022-08-26
> **Response to Reviewer Z2DH**
>
>
> We thank the reviewer for their feedback. We have reorganized Algorithm 1 to be more readable and easier to follow. Additionally, we have completely restructured the Experiments section to be easier to follow. Lastly, we added comparisons to ARCHER and Bias-Corrected HER for the following stochastic environments: TorusFreeze4D (6.2), Stochastic Car (6.3), Red Light (6.4), and Mobile Throwing Robot (6.6).
> We do not claim to outcompete HER or other HER variants on deterministic environments like Fetch -- we only claim that USHER delivers performance comparable to HER, while provably-unbiased methods like DDPG and delta-DDPG do not.
>
> We find that, as expected, neither ARCHER not BHER effectively address HER's bias in stochastic environments. Neither method outperforms HER on any of the environments we tested. By contrast, USHER addresses HER's hindsight bias and achieves better performance on all of stochastic environments we tested.
>
> > For some reason when the authors show the environments in Figure 2, they only show a subset of them. For example, what is 4-Torus with Freeze?
>
> Due to space constraints, we were not able to visualize all of the environments described in the paper. Although we did include explanations of each of the environments in the original paper, we have reorganized the Experiments section so that the explanations of the environments and the results are located together. We have also included a reference in this section to Blier and Olivier (2021), who introduced the environment. Unfortunately, not all the environments can be easily visualized -- 4-Torus is a 4 dimensional torus and therefore does not translate well to paper. Instead, in the supplementary material we have expanded the explanation of each environment that did not receive a visualization in the main body and included visuals where possible.  For 4-Torus with Freeze, we have also included a 2-dimensional torus as a reference point.
>
> We have additionally fixed the spacing error you mentioned.
>
> With respect to $C > 1$, this was not an error. At $c=0$, the importance sampling weight is clipped to exactly 1, which minimizes variance, but reduces to HER and suffers the same biases. As $c \to \infty$, less clipping occurs and the algorithm becomes unbiased, at the cost of greater variance. We found that the bias induced by clipping was neglible for $c>1$
>
>
> We have uploaded the updated version of the paper and supplementary materials.

---

### Official Review · Reviewer_d9ww · 2022-08-01

**Originality:** Very Good
**Technical Quality:** Excellent
**Clarity Of Presentation:** Very Good
**Impact:** 3

**Recommendation:**

Strong Accept: I recommend accepting the paper and will argue for my recommendation even if other reviewers hold a different opinion.

**Summary:**

This paper presents an algorithm to correct for the bias that’s introduced by the Hindsight Experience Replay (HER) algorithm for multi-goal / goal-conditioned RL. The proposed strategy is to separate out the goal used for the reward function from the goal passed to the policy. This allows the derivation of an unbiased variant of HER, using importance sampling. Practical application requires learning the future goal distribution, which is predicted as a separate head off of the existing Q network. Experiments show that this method performs significantly better than HER, as well as other previous unbiased variants of HER, specifically in stochastic environments crafted with HER’s weaknesses in mind. The experiments also show that the importance sampling doesn’t affect HER’s performance in deterministic environments, where bias isn’t a problem.


**Issues:**

It would be nice if you used a consistent color for HER across all graphs (I think it’s different just for Fig 3a and 3b).
In some of the graphs where two of the curves are coincident, it just looks like one of the curves is completely missing. Some correction for this would be useful.


**Quality Of The Limitations Section:**

Limitations are addressed clearly

**Reviewer Expertise:**

4: The reviewer is confident but not absolutely certain that the evaluation is correct

**Robotics Focus:**

Highly relevant to robotics but no hardware experiments

**Strengths And Weaknesses:**

Strengths:
- Paper is clear and the language is easy to read, although it looks like a few passages of text and equations may have compressed a bit due to space limitations.
- The theory is original and novel, and experimental results back up the claims well.

Weaknesses:
- The “transfer” method used for real-robot application is to roll out trajectories in simulation, and then use the same sequence of actions on the real robot applied open-loop. Since the scenario being tested is fairly straightforward (i.e. it’s about picking one risky lane vs another safe one), it’s probably fine, but it doesn’t reflect how a policy might be used in practice.
- How often does one encounter truly stochastic environments in robot applications? (I don’t claim to know the answer to this). Do you think this method will be equally helpful in cases of partial observability?


**Summary Of Recommendation:**

The paper proposes a novel method for unbiased HER in stochastic environments using importance sampling, and experimental evaluations attest to its effectiveness. It’s unclear how often this method will show a benefit in situations encountered in real-world policy learning, but the theory and algorithm are nonetheless a useful contribution to robot learning. I would accept this paper.

---

> ### Author Response · Authors · 2022-08-26
> **Response to Reviewer d9ww**
>
> We thank the reviewer for their feedback and positive review.
>
> > It would be nice if you used a consistent color for HER across all graphs (I think it’s different just for Fig 3a and 3b). In some of the graphs where two of the curves are coincident, it just looks like one of the curves is completely missing. Some correction for this would be useful.
>
>
> Thank you for catching this. We have fixed the color problem in the new version of the paper.
>
> > How often does one encounter truly stochastic environments in robot applications? (I don’t claim to know the answer to this). Do you think this method will be equally helpful in cases of partial observability?
>
> In our experience, action stochasticity is a very common factor in both locomotion and contact-rich environments. This is especially true when single time-steps correspond to longer times in the real world, as it means that small variances have more time to compound.
>
> As for partial observability, we would like to direct the reviewer's attention to the Throwing and Mechanum experiments (Note the simulated Mechanum experiments are now in the appendix). Both of these environments are partially observable. In the Throwing experiment, the wind direction and velocity are sampled at the beginning of the episode and then remain fixed for the rest of the episode. In the Mechanum and physical robot experiments, whether the short path is blocked is sampled at the beginning of the episode and then remains fixed. Neither the wind nor whether the path is blocked are directly observable, so both environments are POMDPs. While USHER is not guaranteed to be unbiased here, we find that in both environments, USHER is significantly less biased than HER, and does not exhibit the same systematic over-optimism that HER does.
>
> Additionally, there exist methods extant in the literature for turning POMDPs into MDPs. One solution is to use a recurrent network on the history, so that the state valuations for the current state are conditioned on all previous observations [1]. Another alternative is belief state planning [2] which can be integrated with neural networks through several means [3] [4]
> for longer time-steps. It should be simple to use one of these methods in conjunction with USHER to get a Markovian state description, which would then guarantee unbiasedness.
>
> [1] Hausknecht, M., & Stone, P. (2017, January 11). Deep Recurrent Q-Learning for Partially Observable MDPs. Association for the Advancement of Artificial Intelligence. https://arxiv.org/abs/1507.06527
>
> [2] Kaelbling, L. P., Littman, M. L., & Cassandra, A. R. (1998). Planning and acting in partially observable stochastic domains. Artificial Intelligence, 101(1–2), 99–134. https://doi.org/10.1016/S0004-3702(98)00023-X
>
> [3] Igl, M., Zintgraf, L., Le, T. A., Wood, F., & Whiteson, S. (2018). Deep variational reinforcement learning for pomdps. Proceedings of the 35th International Conference on Machine Learning, 2117–2126. https://proceedings.mlr.press/v80/igl18a.html
>
> [4] Wang, Y., & Tan, X. (2021). Deep recurrent belief propagation network for pomdps. Proceedings of the AAAI Conference on Artificial Intelligence, 35(11), 10236–10244. https://ojs.aaai.org/index.php/AAAI/article/view/17227

---

### Official Review · Reviewer_KeLa · 2022-08-01

**Originality:** Good
**Technical Quality:** Good
**Clarity Of Presentation:** Good
**Impact:** 3

**Recommendation:**

Weak Reject: I recommend rejecting the paper, but will not argue for my recommendation if the majority of other reviewers have a different opinion.

**Summary:**

This work addresses the “survivorship” bias present in Hindsight Experience Replay when in stochastic environments. In particular, the proposed method USHER estimates an importance weight to de-bias HER. USHER is then evaluated on various simulated robot environments, and the policy learned in one simulated environment is deployed on a physical robot.

**Issues:**

There are a few hyperparameters for USHER like $\alpha_Q$, $\alpha_f$, and the clipping parameter. How are these chosen? How sensitive is USHER to these parameters? Does a clipping parameter of 0.3 work for all environments presented? These details appear to be missing from the main text and appendix, except for the Mechanum robot experiment which appears to use a clipping parameter of 100.

**Quality Of The Limitations Section:**

Limitations are addressed clearly

**Reviewer Expertise:**

3: The reviewer is fairly confident that the evaluation is correct

**Robotics Focus:**

Sufficient demonstration on hardware

**Strengths And Weaknesses:**

Strengths: This work tackles an issue of HER in stochastic environments, and fixes it with importance sampling. To minimize the variance often associated with importance sampling, they simply clip the weights. The proposed method USHER is evaluated on a diverse set of experimental tasks, including discrete/continuous action spaces, deterministic/stochastic environments, and deployment on a physical robot.

Weaknesses: My main concerns are on the experimental validation. Some of the comparisons are missing in some of the performance plots. For example, delta-DDPG and DDPG are missing from the Fetch Push/Slide and Throwing tasks. Also, in these plots, the x-axis is “Epochs,” but it’s not clear how this translates to the number of environment steps. From the experiments performed by Blier and Ollivier, it appears that delta-DQN and delta-PPO are able to achieve a near perfect success rate on FetchReach after about 250 and 1500 trajectories respectively. How do these translate into epochs?

There also appears to be multiple relevant algorithms that attempt to address bias in HER mentioned in the related work, which should be compared to, such as ARCHER, Bias-reduced Multi-step HER, and Bias-corrected HER. Even if these are tackling other forms of bias or incorrectly address bias, I believe they are relevant to compare to.

**Summary Of Recommendation:**

This work addresses the bias of HER in stochastic environments through importance sampling. This method in theory should address the bias, but from the experimental evaluation, it’s difficult to gauge the advantages of USHER over existing methods. In particular, I believe the work can be improved in the following areas: clarifying the x-axis in the plots, inclusion of all comparisons in all of the experimental tasks, comparisons to additional methods, and sensitivity analysis for hyperparameters.

---

> ### Author Response · Authors · 2022-08-26
> **Response to Reviewer KeLa**
>
> **Comment:**
>
> We thank the reviewer for their feedback and suggestions. We have made the following changes at your suggestion
> 1. We added additional evaluations for DDPG and delta-DDPG for the remaining Fetch environments.
> 2. We re-did all relevant plots in terms of the number of episodes instead of the number of epochs.
> 3. We added a hyperparameter analysis to the appendix
> 4. We added comparisons to ARCHER and BHER for the following stochastic environments: TorusFreeze4D (6.2), Stochastic Car (6.3), Red Light (6.4), and Mobile Throwing Robot (6.6). We do not claim to outcompete HER or other HER variants on deterministic environments -- we only claim that USHER delivers performance comparable to HER, while provably-unbiased methods like DDPG and delta-DDPG do not.
>
> Additionally, we noticed a minor typo in the related works section that lead to confusion. In the related works, we said that Bias-reduced Multi-step HER (MHER) attempted to address HER's off-policy bias. This was an error. MHER proposes a variant of HER called Multi-Step HER, and addresses the N-step off-policy bias of Multi-Step HER. It does not attempt to address any form of bias in the standard HER algorithm, and their proposed method reduces to standard HER in the single-step case. Since multi-step algorithms are outside the purview of this work, we do not attempt to compare to MHER.
>
> We find that both ARCHER and BHER fail to address HER's bias in stochastic environments. Neither outperforms HER on any of the stochastic environments we tested. By contrast, USHER addresses HER's hindsight bias and achieves better performance on all stochastic environments.
>
> >From the experiments performed by Blier and Ollivier, it appears that delta-DQN and delta-PPO are able to achieve a near perfect success rate on FetchReach after about 250 and 1500 trajectories respectively. How do these translate into epochs?
>
>
> We adjusted the settings for FetchReach so that the convergence is more visible, in order to answer this question better. We found that USHER and HER achieved almost a 100% success rate after 100 episodes. This makes HER and USHER roughly 2.5X more data efficient than delta-DQN and roughly 150X more data efficient than delta-PPO.
>
>
>
> >There are a few hyperparameters for USHER like $\alpha_Q$, $\alpha_f$, and the clipping parameter. How are these chosen? How sensitive is USHER to these parameters? Does a clipping parameter of 0.3 work for all environments presented? These details appear to be missing from the main text and appendix, except for the Mechanum robot experiment which appears to use a clipping parameter of 100.
>
>
> We have added a section to the appendix examining these hyperparameters. All environments except for TorusFreeze4D and Mechanum use c=0.3. TorusFreeze4D uses c=100 (which is high enough that it functionally means "no clipping") because it was designed as a pathological counterexample for HER, and has much higher hindsight bias than any realistic/natural environment would. Mechanum uses c=100 because we found that there was non-trivial bias when training it with c=0.3. In general, we use c=0.3 as a first pass for each environment, and if non-trivial bias persists we raise c to the point where there is effectively no clipping.
>
> We performed our hyperparameter study by training USHER on FetchReach for 30 episodes with a variety of values for $\alpha_f$ and $\alpha_Q$. The hyperparameter study shows that USHER does not seem to be very sensitive to $\alpha_Q$ as long as it is in the range [0.1, 0.5]. USHER does not seem to be sensitive to the value of $\alpha_f$ at all as long as $\alpha_f \neq 0$ and $\alpha_f \neq 1$.
>
>
> No hyperparameter optimization was performed for $\alpha_Q$ or $\alpha_f$. We guessed an initial value of $\alpha_Q=0.01$ and $\alpha_f=0.5$, got decent results, and never changed them. There is nothing special about these values, except than that both must be greater than 0 for theoretical reasons, $\alpha_f$ should be less than 1 to ensure f does not diverge, and smaller values of $\alpha_Q$ place higher emphasis on hindsight goals, which are more data-efficient than randomly selected goals.
>
>
> We have uploaded the updated version of the paper and supplementary materials.
>
> **Zip File:**
>
> /attachment/2892b8e19eb14f5604afd3cbd3c48f02d9593a69.zip

---

### Meta-Review · Area_Chair_H4Mh · 2022-08-14

**Recommendation:** Accept (Poster)
**Confidence:** 4

**Metareview:**

The paper proposes an important sampling-based technique to correct the over-estimation bias of hindsight experience replay (HER) in stochastic environments. The paper addresses an important problem, and the reviewers have appreciated the contribution. The method mitigates over-estimation in stochastic environments while preserving performance in deterministic environments.

Some of the concerns are:

- Fit to CoRL: The paper is not tackling a robotics problem, but a RL problem that has been applied to a robot to show that the RL algorithm works. The work does not meaningfully advance a robotics problem.

- Missing basline comparison pointed out by the reviewers:  ARCHER, Bias-reduced Multi-step HER, and Bias-corrected HER. delta-DDPG and DDPG are missing from the Fetch Push/Slide and Throwing tasks.

- Paper presentation needs substantial improvement: Algorithm 1 is hard to parse, Figure 3 is confusing -- I would suggest making row/columns consistent and other issues pointed out by Z2DH.

I hope authors address these along with other concerns of the reviewers in the rebuttal.

**Post Rebuttal Update**
The authors have satisfactorily addressed the concerns raised by the reviewers. I believe this paper is a useful addition to goal-conditioned learning that has been demonstrated with real-world robotic experiments.

---

> ### Author Response · Authors · 2022-08-27
> **Response to Area Chair H4Mh**
>
> We thank the area chair and the reviewers for their assessments. We have uploaded a new version of the paper that restructures algorithm 1 and makes the experiment section easier to follow. We have additionally included comparisons to DDPG and delta-DDPG on the Fetch environments, and comparisons to ARCHER and Bias-corrected HER on TorusFreeze4D (6.2), Stochastic Car (6.3), Red Light (6.4), Fetch (6.5), Mobile Throwing Robot (6.6). As we discussed with Reviewers KeLa and Z2DH, we do not claim to outperform any of the other HER variants on deterministic environments like Fetch -- we claim only that USHER's performance is competitive with that of HER, while other unbiased methods like DDPG and delta-DDPG are not. We find that USHER outperforms DDPG and delta-DDPG on all Fetch environments, and ARCHER and Bias-corrected HER on all stochastic environments. As we discussed with Reviewer Kela, Bias-reduced Multi-step HER does not attempt to address HER's bias, but instead addresses the bias of a variant of HER, Multi-step HER. Since multi-step RL is outside the scope of the current work and Bias-reduced Multi-step HER reduces to standard HER in the single-step case, we do not compare to Bias-reduced Multi-step HER.
>
> With regards to our paper's fit for CoRL, we worry we may not have communicated clearly what our contributions are, and how these specifically relate to robotics. Firstly however, we would like to note that HER was not designed as a general purpose RL algorithm, but was specifically intended to solve problems related to reinforcement learning with sparse rewards in high-dimensional robotic object-manipulation environments. The git repo that houses the benchmarks for HER-style algorithms is called Gym-Robotics (https://github.com/Farama-Foundation/Gym-Robotics).
>
> In this work we make three main contributions related to robot learning.
> 1) We demonstrate that HER leads to systematically risk-seeking behavior in realistic robotics scenarios.
> 2) We derive an expression for this bias and provide an asymptotically unbiased algorithm that corrects for it.
> 3) We empirically demonstrate that this algorithm successfully corrects for HER's bias and avoids unecessarily risky behavior on a variety of stochastic robotics tasks, both simulated and real.
>
>
> Prior to our work, it was known that HER was biased in stochastic environments, however it was unclear what behavior this bias would result in for realistic environments. For instance, Blier and Olivier (2021) propose an environment in which HER is asymptotically biased, but this example involves teleporting and non-Euclidean geometry, so tells us little about how HER's bias will manifest in practical cases [1]. We show that in a number of realistic robotics examples, HER underestimates risk. This leads to systematically risk-seeking behavior that has significant safety implications many real robotics scenarios, such as self-driving cars and human-robot interaction. For instance, Li et al (2021) propose using HER for human-aware navigation [2]. Given that humans are highly unpredictable, this bias has significant safety implications for this kind of research, and an analysis of this bias and how to prevent it may be great interest to researchers pursuing similar approaches.
>
>
> The call for papers invites, among other kinds of results, theoretical results relevant to robot learning, and states that papers must demonstrate their relevance to robot learning "through Intent—explicitly address a learning question for physical robots, or Outcome—test the proposed learning solution on physical robots". Contributions 1) and 2) both deliver theoretical results relevant to robot learning. They show that an algorithm designed for deployment on physical robots underestimates risk and can be potentially dangerous, and then derive a solution that avoids these dangerous outcomes. These answer learning questions for physical robots, namely "what kind of behavior does HER's bias induce in physical robots?" and "When is it safe to deploy HER on physical robots?"
> Contribution 3) tests on several robot simulations as well as testing on a real robot, which satisfies the criteria of Robot Learning through Outcome as well.
>
>
> [1] L. Blier and Y. Ollivier. Unbiased Methods for Multi-Goal Reinforcement Learning.
> arXiv:2106.08863 [cs], June 2021
>
> [2] K. Li, Y Lu, and M. Meng. Human-Aware Robot Navigation via Reinforcement Learning with Hindsight Experience Replay and Curriculum Learning  arxiv:2110.04564 [cs], October 2021